# 3D Printed PCB Microfluidics

**DOI:** 10.3390/mi13030470

**Published:** 2022-03-19

**Authors:** Stefan Gassmann, Sathurja Jegatheeswaran, Till Schleifer, Hesam Arbabi, Helmut Schütte

**Affiliations:** Department of Engineering, Jade University of Applied Sciences, 26389 Wilhelmshaven, Germany; sathurja.jegatheeswaran@jstudent.ade-hs.de (S.J.); till.schleifer@student.jade-hs.de (T.S.); hesam.arbabi@student.jade-hs.de (H.A.); helmut.schuette@jade-hs.de (H.S.)

**Keywords:** microfluidics, PCB, rapid prototyping, 3D printing, PCB-MEMS

## Abstract

The combination of printed circuit boards (PCB) and microfluidics has many advantages. The combination of electrodes, sensors and electronics is needed for almost all microfluidic systems. Using PCBs as a substrate, this integration is intrinsic. Additive manufacturing has become a widely used technique in industry, research and by hobbyists. One very promising rapid prototype technique is vat polymerization with an LCD as mask, also known as masked stereolithography (mSLA). These printers are available with resolutions down to 35 µm, and they are affordable. In this paper, a technology is described which creates microfluidics on a PCB substrate using an mSLA printer. All steps of the production process can be carried out with commercially available printers and resins: this includes the structuring of the copper layer of the PCB and the buildup of the channel layer on top of the PCB. Copper trace dimensions down to 100 µm and channel dimensions of 800 µm are feasible. The described technology is a low-cost solution for combining PCBs and microfluidics.

## 1. Introduction

Microfluidics has a lot of advantages. The usage of low volumes, the fast and well-controlled chemical reactions, the portability and the effects that appear at the micro-scale meet the needs of a huge range of applications. At the beginning of microfluidics research, a sophisticated micro-technologies lab was needed. The main technologies of the semi-conductor industry were used to generate the needed cavities. During the course of microfluidic technology development, many other technologies were developed, such as micro-machining, soft lithography, etc. [1,2,3]. One technological possibility combines printed circuit boards (PCBs) and micro-channels [4]. The advantage of this approach is the easy combination of electrodes, electronics and fluidic channels, as well as the well-established mass production of PCBs. The combination of electrodes and fluidics is of interest in all applications where an electric connection to the fluid is needed, e.g., electrochemical measurements, impedance spectroscopy or conductivity measurements. The combination of electronics and fluidics is also of interest in all applications where sensors and actuators that have electric connections need to be in contact with the fluid (e.g., thermal actuators by joule heating, electromagnetic, electrostatic, electrokinetic and electrowetting actuators, photometric sensors with LEDs and photo diodes and many more).

Additive manufacturing, also known as 3D printing, has also been used for the creation of microfluidics [5]. One very promising technology is masked stereolithography (mSLA), where photosensitive resins are polymerized by light activation. Many different resins with different properties (even biocompatibility) are on the market. A lot of affordable mSLA printers are available with typical x/y resolutions down to 35 µm. The most advanced microfluidic use of resin-based 3D printing is presented by Gong et al. Using customized hardware and resin, 18 µm × 20 µm-sized cavities are feasible [6].

In order to facilitate the fabrication of, and to disseminate the advantages of, microfluidics, the usage of commercially available mSLA printers and resins is of interest. This will enable the creation of microfluidic devices with low investments in lab infrastructure.

The combination of 3D printing and PCBs for fluidics was reported in [7]. Cabrera-López et al. used a 3D printed well and a commercially produced PCB for impedance spectroscopy measurements. The 3D printed part was made by an FDM (fused deposition modelling) printer, and contains no closed cavities. Both parts were detachable.

In this paper, a technology for the creation of microfluidics on PCB substrates using commercially available resins and mSLA printers is shown. The demonstrated technology includes the structuring of the copper layer of the PCB and the buildup of a channel layer, all using a commercially available mSLA printer. This technology can be used to create microfluidic designs in a simple laboratory environment in a short time. In time-limited education events such as summer schools the technique presented here will enable students to build their own designs.

In this paper, a microfluidic chip for the measurement of salinity using conductivity measurement is demonstrated. The chip contains a water-filled cavity and electrodes to perform the conductivity measurement. The production process and the first results of the chip test are demonstrated.

## 2. Materials and Methods

First the technology of the combination of PCBs and microfluidic channels is described. Second the example system that is created is shown in detail.

### 2.1. Technology

The 3D printer used here is a Sonic Mini 4k (Phrozen, phrozen3d.com). It is a commercially available mSLA printer with a resolution of 35 µm in x- and y-, and 10 µm in the z-direction. The build area is 13.4 cm × 7.5 cm × 13 cm. Other printer models can also be used. The only modification made was a holder for PCBs with the size of 50 mm × 50 mm.

Figure 1 shows the build plate of the used 3D printer with a mounted PCB. An aluminum frame with alignment points (highlighted in Figure 1) is added to the build plate. The alignment points are circular-shaped touchpoints created by milling in the opening for the PCB. With this manual alignment, an accuracy of 100 µm is feasible.

The PCB holder is the only modification needed to use the technology presented here.

All technology steps are pictured in Figure 2. The technology starts with the structuring of the copper (Figure 2a). For this, a single-sided PCB (1.5 mm FR4, 35 µm copper), coated with photosensitive resist, is used (presensitized boards, Bungard Elektronik & Co KG), and is cut into 50 mm × 50 mm pieces. The PCB is mounted with double-sided stick tape in the holder (Figure 1); after this, the *Z*-axis calibration takes place. As a printing file, the negative layout of the copper traces is loaded to the printer. The resist film needs to be exposed for 20 min with the used printer. After this, the exposed layer needs to be developed in the Bungard developer (article number: 72130–01, Bungard Elektronik & Co KG) for 5 min. The mask for etching the copper is now ready. For etching, a solution of sodium persulfate (ETCHANT 400 g, PROMA) can be used. The PCB used here was etched in an etching machine (AETZGERAET1, PROMA) at 40 °C for 10 min. The proposed procedure leads to the best results, with copper trace width down to 100 µm. The usage of the 3D printing resin as etch resist was also tested. It worked down to a copper trace width of 300 µm.

The second step is the printing of the channel layer on the structured PCB (Figure 2b). Right before printing the channel layer, the PCB must be cleaned with isopropanol. The so-prepared PCB is mounted again in the holder on the printer build plate using double-sided sticky tape. The same orientation as for the exposure of the etch resist should be used. This ensures the alignment of the copper trace with the channels. In preparation, before the print can start, *Z*-axis calibration must be performed with the mounted PCB. Now the channel layer can be printed directly on the PCB. For this purpose, the vat must be installed and filled with the resin. In addition, the printer file for the channel structure must be transferred on the printer and selected for printing. In this study, the color mix resin basic from 3Djake (niceshops GmbH, Paldau, Austria) was used. This resin needs longer exposure times than other resins; however, this longer exposure time leads to less light bleed and less over-exposure for the base layers, which makes it suitable for printing small cavities. The settings are: 4 bottom layers, 80 s exposure time for the bottom layer, 6.5 s exposure time for normal layers. The *z*-axis resolution used was 50 µm. As usual for the mSLA, 3D printing of the layer-by-layer buildup of the designed structure takes place at this step.

After printing the channel structures, the print must be cleaned (Figure 2c). All uncured resin must be washed off before final curing. For this purpose, the wash-and-clean station CW1S (Prusa Research a.s., Prague, Czech republic) was used. Cleaning was performed in isopropanol for 3 min. After the outside cleaning, the remaining resin inside the cavities must be removed. For this, a syringe filled with isopropanol was used, followed by cleaning with pressurized air. After cleaning, the final curing takes place, also in the CW1S. The curing program was run for 3 min.

The total procedure can be performed in less than three hours. This makes this technology suitable for time-limited education courses.

### 2.2. Example (Salinity Sensor)

The usage of PCBs as substrate for microfluidics has the advantage that the integration of electrodes and electronics in the microfluidic system is intrinsic. Using this approach, chip electrophoresis, conductivity measurements, spectrophotometry and many other applications can be built. Here, the conductivity measurement of seawater for the measurement of salinity is chosen as an example.

The system was designed using the 3D CAD program Fusion360^®^ (Autodesk Inc.), which gives the possibility of co-designing the PCB and the channel. This has the advantage that the positioning of the electrodes and electronics and the channels can be designed properly in one tool. The steps for an electro-mechanical co-design are described in the Fusion360^®^ documentation [8]. Figure 3 shows the CAD view and one realized prototype.

The example used here consists of 5 different channels with different electrode arrangements for measuring the conductivity of the water in the channels. The different arrangements are for the demonstration of the 2-probe, the 4-probe and the guard electrode measurement. Closed cavities are useful in conductivity measurements to build a well-defined current path and prevent external fields. The system was built to demonstrate the salinity measurement to marine science students.

In the experiment, artificial seawater was created using sea salt and de-ionized water. The conductivity of the water was 55.3 mS/cm; it was measured with a GMH 3400 conductivity meter (Greisinger Messtechnik GmbH). The measurement of the resistance of the seawater in the channels was performed by an LCR meter HM8018 (Hameg instruments) at 25 kHz measurement frequency. The seawater was filled in the channels by 1 mL plastic syringes.

## 3. Results

### 3.1. Technology Results

Using the described technology systems containing electrodes, electronics and microfluidic channels are feasible. Only commercially available tools and consumables were used. The structure sizes were measured with a measuring microscope prior to the functional tests of the system. The minimal structure sizes of the 35 µm-thick copper traces are 100 µm. The cavity width (x/y resolution of the printer) in the presented system was 800 µm, which is appropriate for the demonstrated example but should be reduced in the future. Gong et al. concluded that a cavity dimension of 4 times the x/y resolution (pixel size) is feasible [6]. This is only possible using customized hardware and resin. The height of the cavity (*z*-direction of the printer) was measured by filling the cavity with a defined volume of water. Assuming a rectangular shape of the cavity, heights of about 200 µm were calculated. This was one-fifth of the designed height. The reason is that the entrapped resin in the cavity is exposed to a light dose with every layer above the cavity. Finally, this leads to an unwanted polymerization of the upper layer in the cavity. This effect should be taken into account when designing the cavity height.

An influence of the different optical properties of the copper and the FR4 was not observed during the production of the structure presented here. It may become apparent when the feature size of the channels becomes smaller and the adjacent copper areas become larger.

The problem of creating cavities with mSLA printers is that the enclosed resin that should stay not polymerized becomes exposed by stray light (this effect is also named light bleeding). This effect depends on the following factors: 1. Polymerization depth of the resin. A good description of the effects leading to this behavior can be found in [9]; 2. Design. Lower number of layers above the cavity and bigger inner dimensions of the cavity overcome the problem of polymerized encapsulated resin.

### 3.2. Conductivity Measurement Results

The results for the conductivity measurement are shown here only for the four-probe channel; the measurements of the other electrode arrangements agree with the results presented here.

With the calculated cross-section of the four-probe channel and the conductivity of the salt water, the resistance was calculated as 19.9 kOhm. The measured value with the HM 8018 was 20.9 kOhm (@25 kHz). The reason for the difference in the values could be the non-rectangular cross-section of the channel.

Using this microfluidic chip, the principle of salinity measurement by direct contact with the water can be demonstrated. The difference in electrode arrangements, the influence of the outside electric path and the usage of a guard electrode can be shown. With the system shown here, students will gain experience with the electric current field and with measuring salinity.

## 4. Conclusions

Microfluidics is no longer a domain of microtechnology laboratories with expensive equipment. If a cavity width of several hundreds of micrometers is enough, it is even possible to build microfluidic systems with a low-cost mSLA 3D printer and commercially available resins. The presented technology shows also that the combination of electrodes, electronics and microfluidic channels can be produced using the low-cost mSLA technique. The novelty of the technology is the usage of a PCB as a substrate. PCBs usually carry electronics and realize their wiring. When using PCBs as a substrate for microfluidics, the combination of electronics and fluidics is intrinsic. Combined electronic/microfluidic systems can integrate the electric connection to the fluid easily. The evaluation electronics can be located close to the sensors, and the system becomes compact and reliable due to the lack of wiring.

The technology shown here is a very good option for rapid prototyping and for educational purposes because of the low investments and the fast results. For mass production of PCB-based microfluidics, other technologies, such as the combination of hot embossed thermoplastic channels with mass-produced PCBs, are suitable.

With this technology, the structuring of the copper clad is possible in structure sizes down to 100 µm. The realized cavity width above the copper layer was 800 µm. Further research was will be carried out to minimize the feasible cavity size.

As an example, a microfluidic conductivity sensor chip was realized. Several electrode arrangements were demonstrated. The principle of salinity measurements using direct contact with the water, different electrode arrangements, such as the four-probe measurement, and the influence of the outside current path were demonstrated using the presented microfluidic chip.

## Figures and Tables

**Figure 1 micromachines-13-00470-f001:**
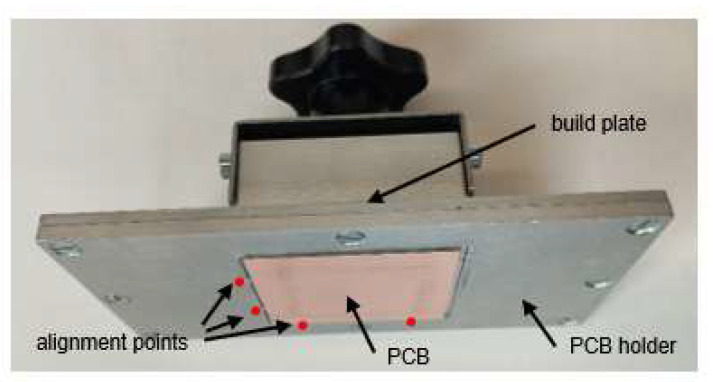
Printer build plate with holder for a PCB; the marked areas are alignment posts.

**Figure 2 micromachines-13-00470-f002:**
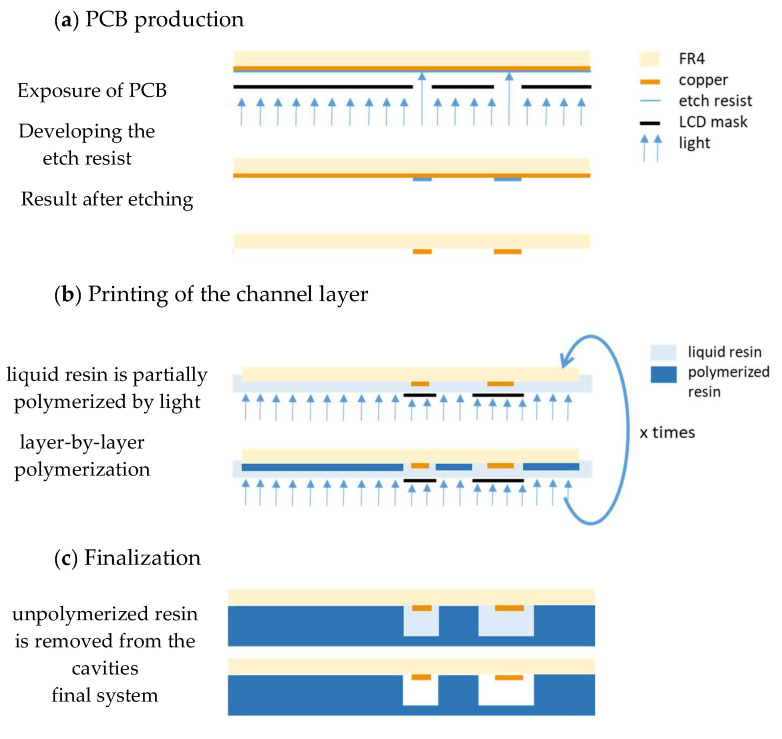
Technology steps: (**a**) PCB production—the etch resist is exposed by the 3D printer after the copper layer is etched as usual; (**b**) printing channel layer—the 3D printing process is carried out on the structured PCB; (**c**) finalization—unpolymerized resin must be removed from the cavities.

**Figure 3 micromachines-13-00470-f003:**
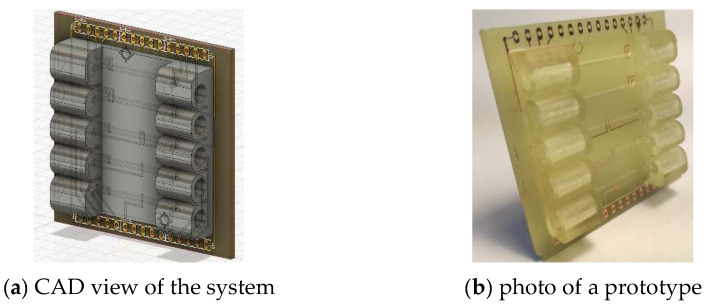
The view in the CAD system and one realized prototype: (**a**) CAD view of the combined design—the PCB with the copper traces is in the back; the channels and the connectors are opaque gray; (**b**) photo of one realized prototype—the channel material is opaque for better visibility of the fluid.

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
