# Peer review of "3D Printed PCB Microfluidics"

_micromachines, 2022, doi:10.3390/mi13030470_

Round 1

Reviewer 1 Report

In this draft, a novel approach to microfluidic fabrication is presented, using additive manufacturing directly on a PCB. All methods and materials are correctly detailed. But some things need to be clarified: 1) There are some alignment points for PCB positioning. What are they? 2) What is the accuracy of the above alignment? 3) PCB substrate has different reflective properties, that is, copper is much more reflective than FR4. How does it affect the performance of the SLA 3D printer? 4) The described manufacturing process is valid for rapid prototyping, but it is not an option for mass production. This should be noted in the conclusions.

Author Response

Dear reviewer,

thank You for Your valuable input to improve my paper.

I will answer here on You questions and will also add these information to the manuscript.

1.) The alignment points are circular shaped touchpoints created by milling in the opening for the PCB.

2.) With this manual alignment an accuracy of 100µm is feasible.

3.) An influence of the different optical properties of the copper and the FR4 was not observed during the production of the here  presented structure. It may become apparent when the feature size of the channels get smaller and the adjacent copper areas become larger.

4.) The technology shown here is a very good option for rapid prototyping and for educational purposes because of the low investments and the fast results. For mass production of PCB based microfluidics other technologies like the combination of hot embossed thermoplastic channels with mass produced PCBs are suitable.

Best regards

Stefan Gassmann

Reviewer 2 Report

The authors describe a method to create a microfluidic device on a PCB substrate using mSLA printer. This is an approach that can be easily and widely implemented for the development of lab-on-chip sensors that combine electrodes and fluidics channels.

Summarized below suggestions to improve the manuscript.

  • Elaborate more on the advantages and specific applications of the described technology.
  • The introduction is lacking sufficient background information on previous work in the field, more specifically 3D printed electrochemically systems.

Dong, Yue, Xin Min, and Woo Soo Kim. "A 3-D-printed integrated PCB-based electrochemical sensor system." IEEE Sensors Journal 18.7 (2018): 2959-2966.

Cabrera-López, John Jairo, et al. "PCB-3D-Printed, reliable and reusable wells for impedance spectroscopy of aqueous solutions." Journal of Physics: Conference Series. Vol. 1272. No. 1. IOP Publishing, 2019.

  • The different components in Figure 1 need to be labeled.

Author Response

Dear reviewer,

thank You very much for Your valuable input to improve my paper.

I will add the suggested improvements to my paper and answer You questions also here.

  1. ) I will add a more about the advantages and the applications of the described technology.
  2. ) Thank You for making me aware of the electrochemical applications of PCBs in combination with 3D printing.  I will add it to my paper. But the given examples are not comparable with the here presented work. Dong et al. printed the electric circuit itself with conductive ink. In there work no channels were used. Cabra-Lopez et al. attached a 3D printed well to a PCB. The connection between the PCB and the well is detachable. This is not the case in the presented work.
  3. I added labels in Fig. 1

Best regards and many Thanks

Stefan Gassmann